# LQG Control for Dynamic Positioning of Floating Caissons Based on the Kalman Filter

**DOI:** 10.3390/s21196496

**Published:** 2021-09-29

**Authors:** Jose Joaquin Sainz, Elías Revestido Herrero, Jose Ramon Llata, Esther Gonzalez-Sarabia, Francisco J. Velasco, Alvaro Rodriguez-Luis, Sergio Fernandez-Ruano, Raul Guanche

**Affiliations:** 1Department of Electronic Technology, Systems Engineering and Automatic Control, E.T.S. de Ingenieros Industriales y de Telecomunicacion, University of Cantabria, Av. de los Castros s/n, 39005 Santander, Spain; josejoaquin.sainz@unican.es (J.J.S.); llataj@unican.es (J.R.L.); esther.gonzalezs@unican.es (E.G.-S.); 2Department of Electronic Technology, Systems Engineering and Automatic Control, E.T.S. de Náutica, University of Cantabria, C/Gamazo 1, 39004 Santander, Spain; velascoj@unican.es; 3Environmental Hidraulics Institute, IH Cantabria, Universidad de Cantabria, c/Isabel Torres n 15, Parque Cientifico y Tecnológico de Cantabria, 39011 Santander, Spain; alvaro.rodriguezluis@unican.es (A.R.-L.); sergio.fernandezruano@unican.es (S.F.-R.); raul.guanche@unican.es (R.G.)

**Keywords:** LQG, dynamic positioning, Kalman filter, linear quadratic regulator (LQR)

## Abstract

This paper presents the application of an linear quadratic gaussian (LQG) control strategy for concrete caisson deployment for marine structures. Currently these maneuvers are carried out manually with the risk that this entails. Control systems for these operations with classical regulators have begun to be implemented. They try to reduce risks, but they still need to be optimized due to the complexity of the dynamics involved during the sinking process and the contact with the sea bed. A linear approximation of the dynamic model of the caisson is obtained and an LQG control strategy is implemented based on the Kalman filter (KF). The results of the proposed LQG control strategy are compared to the ones given by a classic controller. It is noted that the proposed system is positioned with greater precision and accuracy, as shown in the different simulations and in the Monte Carlo study. Furthermore, the control efforts are less than with classical regulators. For all the reasons cited above, it is concluded that there is a clear improvement in performance with the control system proposed.

## 1. Introduction

Nowadays, precast concrete (floating) caissons are used in the construction of maritime infrastructures all over the world. These types of constructions have been highly developed in the last few years [1]. Various operations are carried out to complete the construction of a maritime infrastructure, among which the most important are the following: sea transportation from the construction yard to the target site, positioning at the target site and sinking to its final position. Within the previously-mentioned operations, the most important ones are sinking and positioning. The achievement of these operations entails a level of risk. In addition, a small margin of deviation is allowed, less than 10 cm of deviation from theoretical values. This value is provided by the SAFE Project stakeholder, FCC, and is based in [2].

One of the standard procedures is the fine positioning of the caisson before proceeding with the sinking operations. Due to the complexity of the various operations involved during the whole process, the personnel involved in them must be adequately trained for the correct execution of the whole process. Furthermore, the complex operating environment in which the operations are carried out may provoke inaccuracies in the positioning of the caisson or collision with previous caissons. This may cause material damage to the equipment and even damage to the personnel who carry out the operations. Moreover, apart from accidents this maneuver may lead into inaccurate positioning of the caisson, something which may lead to economic losses. The sinking processes, which take place just after the fine positioning, is also a highly complex maneuver where the sinking velocity needs to be adjusted in order to avoid violent seabed contacts without increasing the positioning potential inaccuracies. 

Therefore, and considering all of the above, in this work, we proposed a dynamic positioning (DP) system for anchored caissons, which contributes to the minimization of the operation risks in a way that safeguards the integrity of equipment and people. In addition, and simultaneously, it will make the operations economically less costly due to the reduction of the necessary personnel on board in the course of the operations.

In the state of art relative to vessels or floating structures [3,4], there are several DP approaches, which apply a wide variety of control methods. Among all these methods, the most outstanding are the following: PID control [4,5], acceleration feedback and Kalman filtering [4,6,7,8,9], robust control [10,11,12], Fuzzy control [13], backstepping method [14,15] and others [16,17,18]. Although with some similarities, all the cited references related to floating structures and ships differ from the dynamic behavior of the floating caissons presented in this work. Apart from that, the allocations system used in all the cited contributions is based on thrusters, which is completely different from the one proposed in this work due to being based on mooring lines. Besides, it is found a work where it is used a classical controller for a DP system of caissons [19]. The approach of the present paper differs from this one, since we propose the application of a linear quadratic gaussian (LQG) control strategy in a DP system, which normally demands a compound of staff and equipment when it is performed in the non-automated way. 

As far as the LQG control strategy is concerned, some relevant contributions are found in the literature, related to the motion control of underwater vehicles [20,21,22] or their thrusters [23], to cite some of the most representative. In the same way, regarding to stabilization and motion control of ships, there is a variety of references [24,25,26,27]. All of these contributions include thrusters in the allocation system, see [28] for different types of thruster configurations, or [8] for marine vehicles and [29] for a general approach of control allocation algorithms. It is possible to check in the contributions published by [30,31,32,33,34] that DP control for moored vessels is also performed with thrusters, where the mooring system is modelled by means of finite elements [35].

The novelty of the present paper lies in the first-time application of the LQG control method to DP of floating caissons with an allocation system based on mooring lines. This approach differs from those previously reported in the literature, since it is performed without thrusters. We propose the application of an LQG control strategy based on the Kalman filter (KF) to a DP system of anchored floating caissons, see Figure 1. In this proposal, we implement a control allocation system without thrusters but based on teleoperated mooring lines. The relative position between the caisson and the anchor points changes the direction of the forces induced on the caisson by the different mooring lines. This highlights instability to the system, which constitutes a difficulty that the LQG control strategy must overcome.

Subsequently, the results between the classical controller applied to caissons [19] and the controller proposed in this paper will be compared. It will be observed that the proposed controller provides better results in the actuation signals and a reduction in the oscillations under the same perturbations conditions.

## 2. The Model

### 2.1. Hydrodynamic Model of the Caisson

The dynamic model of the caisson, object of study of this paper is represented by the following Cummins equation [36,37]:(1)$$(M+A\left(\infty \right))+\ddot{z}\left(t\right)+\int_{0}^{t}\bar{K}\left(t-\tau \right)\dot{z}\left(\tau \right)d\tau +Cz\left(t\right)\;=\;F_{a0}\left(t\right)+F_{w0}\left(t\right)$$

The function of delay and fluid memory effects is:(2)$$\bar{K}\left(t\right)\;=\;\frac{2}{\pi }\int_{0}^{\infty }B\left(w\right)cos\left(wt\right)dw$$
where $z\left(t\right)\;=\;{\left[x_{0},y_{0},z_{0},\phi_{0},\theta_{0},\psi_{0}\right]}^{T}$ are the position and the Euler angles of the caisson. *M* is the mass of the caisson, A(∞) is the added mass at infinite frequency, *K* is the function of delay and fluid memory effects, B(w) is the damping coefficient for every frequency, *C* is the hydrostatic restoration coefficient, $F_{a0}\;=\;\left[X_{a0},Y_{a0},Z_{a0},K_{a0},M_{a0},N_{a0}\right]$ are the control forces and moments, $F_{w0}\;=\;\left[X_{w0},Y_{w0},Z_{w0},K_{w0},M_{w0},N_{w0}\right]$ are the forces and moments induced by disturbances.

### 2.2. Linear Approximation

The hydrodynamic model previously presented is nonlinear, see Equation (1). It is well-known that a linear quadratic regulator (LQR) provides a good performance if the dynamic behavior of the system is linear. That is why, in this section, we obtained a linear approximation of the model in Equation (1). Taking into consideration the operational conditions of this paper, which refer to dynamic positioning of caissons at low speed, a linear approximation can properly represent the dynamics of the caisson.

The structure of the linear approximation (Figure 2 and Table 1) is determined based on previous knowledge of the system. To do so, the step response was evaluated by observing fundamental aspects, such as the appropriate sampling period, the natural modes of the system, coupling among the different inputs and outputs of the system, delays, etc. Subsequently, input signals are generated for the model in Equations (1) and (2) based on square wave signals and pseudorandom binary sequence (PRBS) signals that are applied sequentially in each of the inputs of the system, obtaining the outputs response: $x_{0},\;y_{0},\;z_{0},\;\phi_{0},\;\theta_{0}\;\mathrm{and}\;\psi_{0}$.

Subsequently, by means of least squares [38,39,40,41] a family of parametric models is obtained in the form of transfer functions, which best fit to the data, see Figure 2 and Table 1. Then, in order to properly apply the LQG control strategy, the family models are converted into state space representation of Equations (3) and (4).

The cited linear approximation in state space form is:(3)$${\dot{x}}_{L}\left(t\right)\;=\;A_{L}x_{L}\left(t\right)\;+\;B_{L}u_{L}\left(t\right)$$
(4)$$y_{L}\left(t\right)\;=\;C_{L}x_{L}\left(t\right)\;+\;D_{L}u_{L}\left(t\right)$$
$$A_{L}\;=\;\left[\begin{array}{cccccc} 1 & 0 & 0 & 0 & 0 & 0\\ 0 & 1 & 0 & 0 & 0 & 0\\ 0 & 0 & 0 & 0 & 0 & 0\\ 0 & 0 & 0 & 0 & 0 & 0\\ 0 & 0 & 0 & 0 & 0 & 0\\ 0 & 0 & 0 & 0 & 0 & 1 \end{array}\right]$$
$$B_{L}\;=\;\left[\begin{array}{ccc} b_{11} & 0 & 0\\ 0 & b_{22} & 0\\ 0 & 0 & 0\\ 0 & 0 & 0\\ 0 & 0 & 0\\ 0 & 0 & b_{63} \end{array}\right]$$
$$C_{L}\;=\;eye\left(6\right)$$
$$D_{L}\;=\;zeros\left(6,3\right)$$
where: 

$A_{L}{}_{6x6}$ is the state matrix.$x_{L}\left(t\right)\;=\;{\left[x_{L},y_{L},z_{L},\phi_{L},\theta_{L},\psi_{L}\right]}^{T}$ is the state vector. $B_{L}{}_{6x3}$ is the input matrix.$u_{L}\left(t\right)\;=\;{\left[X_{L},Y_{L},N_{L}\right]}^{T}$ is the input vector. $C_{L}{}_{6x6}$ is the output matrix.$y_{L}\left(t\right)\;=\;{\left[x_{L},y_{L},z_{L},\phi_{L},\theta_{L},\psi_{L}\right]}^{T}$ is the output vector. $D_{L}{}_{6x3}$ is the feed through matrix.

Figure 3, exhibits the fit of the obtained linear approximation for the surge degree of freedom with respect to the data generated by the nonlinear model of Equation (1). This model validation is performed, as usual in the application of parameter estimation theory, by using different data than those used in the estimation. As can be seen, the linear approximation fits the response of the nonlinear model perfectly. There are no discrepancies between the two responses. This indicates that, for the present operating conditions, the linear approximation system can be used.

As previously discussed, due to the low speed at which the operations are performed, the linear approximation correctly represents the system dynamics. The proposed controller works correctly for linear systems. In case the operations were performed at a higher speed, it is possible that this linear approximation would not be valid and, therefore, another type of control would have to be implemented. However, floating caissons perform their operations in a slow motion because their own structure and characteristics prevent the operations to be performed at a high speed. As a result of that, the proposed controller can be considered to be fully valid and applicable to the system for which it has been designed.

### 2.3. Model of Wave Disturbances

According to [3] the effects of the waves can be divided in two effects:

The effects of the first-order waves ($w_{F}$). The effects of first-order waves are small oscillations of zero mean.The effects of the second-order waves (LF). The effect of second-order wave are typically represented by slow drift motions. 

The following state space representation can be used to represent the first-order wave effects:(5)$${\dot{x}}_{w}\left(t\right)\;=\;\left[\begin{array}{ccc} -2\lambda w_{0} & 0 & 0\\ 0 & -2\lambda w_{0} & 0\\ 0 & 0 & -2\lambda w_{0}\\ 1 & 0 & 0\\ 0 & 1 & 0\\ 0 & 0 & 1 \end{array}\begin{array}{ccc} -w_{0}{}^{2} & 0 & 0\\ 0 & -w_{0}{}^{2} & 0\\ 0 & 0 & -w_{0}{}^{2}\\ 0 & 0 & 0\\ 0 & 0 & 0\\ 0 & 0 & 0 \end{array}\right]x_{w}\left(t\right)+\left[\begin{array}{c} 2\lambda \omega_{0}\sigma \\ 2\lambda \omega_{0}\sigma \\ 2\lambda \omega_{0}\sigma \\ 0\\ 0\\ 0 \end{array}\right]u_{w}\left(t\right)$$
(6)$$y_{w}\left(t\right)\;=\;\left[\begin{array}{c} Identity_{3x3}\\ 0_{3x3} \end{array}\right]x_{w}\left(t\right),$$
where λ is the damping, the gain $K_{\omega }\;$ is $K_{\omega }$ = $2\lambda \omega_{0}\sigma$, σ is the wave intensity, $\omega_{0}$ is the wave dominant frecuency and w is the Gaussian white noise, the state vector is $x_{w}\left(t\right)\;=\;{\left[x_{w},y_{w},z_{w},\varphi_{w},\theta_{w},\psi_{w}\right]}^{T}$ and the input vector is $u_{w}\left(t\right)\;=\;{\left[w_{x},w_{y},w_{\psi },0,0,0\right]}^{T}$. These effects must be added to each of the outputs of the system to be contaminated.

Moreover, the second-order wave drift forces $F_{w}$ = ${\left[X_{w},Y_{w},N_{w}\right]}^{T}$ are modelled as slow varying bias terms (wiener processes) [3,6]:(7)$${\dot{X}}_{w}\left(t\right){\;=\;w}_{X}$$
(8)$${\dot{Y}}_{w}\left(t\right){\;=\;w}_{Y}$$
(9)$${\dot{N}}_{w}\left(t\right){\;=\;w}_{N},$$
where $w_{X}$, $w_{Y}$ and $w_{N}$ are sequences of white noise.

Then, the complete model used in this work including first and second-order wave effects is:(10)$${\dot{x}}_{L}\left(t\right)\;=\;A_{L}x_{L}\left(t\right)\;+\;B_{L}\left[u_{L}\left(t\right)+F_{w}\left(t\right)\right]\;+\;w\left(t\right)$$
(11)$$y_{L}\left(t\right)\;=\;C_{L}x_{L}\left(t\right)\;+\;D_{L}\left[u_{L}\left(t\right)+F_{w}\left(t\right)\right]$$
(12)$$y_{t}\left(t\right)\;=\;y_{L}\left(t\right)\;+\;y_{w}\left(t\right)+v\left(t\right),$$
where v(t) is the noise caused by the sensors and this effect is modeled with Gaussian white noise, w(t) is the process noise and is also modeled with Gaussian white noise and $y_{t}\left(t\right)$ is the output vector with the addition of first-order waves and sensor noise.

## 3. Kalman Filter

The LQG control strategy of this work is based on the KF. We use this filter to estimate the states and the wave effects previously defined in Equations (5) and (6). It must be noted that only the first-order wave effects are estimated with the KF. The second-order wave effects are compensated with a proportional integral (PI) controller.

In order to estimate and filter, the first-order wave Equations (5) and (6), it is necessary to increase the states of the model with which the KF works, as indicated in [7]. The model with augmented states corresponds to:(13)$${\dot{x}}_{f}\left(t\right)\;=\;\left[A_{an}\right]x_{an}\left(t\right)\;+\;\left[B_{an}\right]u_{an}\left(t\right)\;+\;w\left(t\right)$$
(14)$${\dot{y}}_{f}\left(t\right)\;=\;\left[C_{an}\right]x_{an}\left(t\right)\;+\;v(t)$$
with,
$$x_{f}\left(t\right)\;=\;{\left[x_{L},y_{L},z_{L},\phi_{L},\theta_{L},\psi_{L},x_{w},y_{w},z_{w},\varphi_{w},\theta_{w},\psi_{w}\right]}^{T}$$
$$y_{f}\left(t\right)\;=\;{\left[x_{f},y_{f},\psi_{f}\right]}^{T}$$
$$u_{an}\left(t\right)\;=\;\left[\begin{array}{c} u_{L}\left(t\right)\\ u_{w}\left(t\right) \end{array}\right];$$
$$A_{an}\;=\;\left[\begin{array}{c} A_{L6x6}\\ 0_{6x6} \end{array}\begin{array}{c} \;0_{6x6}\\ \;A_{w}{}_{6x6\;} \end{array}\right];$$
$$B_{an}=\left[\begin{array}{cc} \;B_{L6x3} & 0_{6x3\;}\\ \;0_{6x3\;} & B_{w}{}_{6x3\;} \end{array}\right];$$
$$C_{an}=\left[Identity_{6x6\;}\begin{array}{cccccc} 1 & 0 & 0 & 0 & 0 & 0\\ 0 & 1 & 0 & 0 & 0 & 0\\ 0 & 0 & 0 & 0 & 0 & 0\\ 0 & 0 & 0 & 0 & 0 & 0\\ 0 & 0 & 0 & 0 & 0 & 0\\ 0 & 0 & 0 & 0 & 0 & 1 \end{array}\;\right];$$

$u_{an}$(*t*) are the inputs of the caisson and $u_{w}\left(t\right)$ is the wave input vector, $y_{f}\left(t\right)$ is the KF output vector, v(t) is the noise caused by the sensors and this effect is modeled with Gaussian white noise and w(t) is the process noise and also is modeled with Gaussian white noise. This process noise is incorporated to take into account the discrepancy between the model approximation and real dynamic of the caisson.

The algorithm used in the implementation of a KF as indicated in [3,8] is:

The matrices design
(15)$$Q\left(t\right)\;=\;Q^{T}\;>\;0$$
(16)$$R\left(t\right)\;=\;R^{T}\;>\;0$$
(17)$${\hat{x}}_{f}\left(0\right)\;=\;x_{f}{}_{0}$$Initial conditions
(18)$$P\left(0\right)\;=\;E\left[\left(x_{f}\left(0\right)-{\hat{x}}_{f}\left(0\right)\right){\left(x_{f}\left(0\right)-{\hat{x}}_{f}\left(0\right)\right)}^{T}\right]\;=\;P_{0}$$The Kalman profit matrix propagations
(19)$$K\left(t\right)\;=\;P\left(t\right)H^{T}\left(t\right)R^{-1}\left(t\right)$$*P* is the solution of the Riccati algebraic equationPropagation of the estimated state
(20)$${\dot{\hat{x}}}_{f}\left(t\right)\;=\;A_{an}\left(t\right){\hat{x}}_{f}\left(t\right)+B_{an}\left(t\right)u_{an}\left(t\right)+K\left(t\right)\left[y_{f}\left(t\right)-H\left(t\right){\hat{x}}_{f}\left(t\right)\right]$$

The tuning parameters are the covariance matrix of the noise *R* and the covariance matrix of the states *Q*. The tuning of the KF is done through the entries of the state and measurement noise [8]. If the model is believed to be uncertain, we need to increase the state covariance.

The noise from the sensors was considered to be uncorrelated. Therefore, *R* was implemented as the diagonal of the sensor noise covariances.
(21)$$R\;=\;diagonal\left(\sigma_{x},\sigma_{y},\sigma_{z},\sigma_{\phi },\sigma_{\theta },\sigma_{\psi }\right).$$

The state covariance matrix was also established diagonally for the same considerations
(22)$$Q\;=\;diagonal\left(Q_{x},Q_{y},Q_{z},Q_{\phi },Q_{\theta },Q_{\psi }\right).$$

## 4. LQG Control Strategy

The LQG control is derived from an LQR controller, the difference between them is that the state feedback in an LQG is obtained by the KF [19]. In this work, we use the KF for the states and waves estimation as indicated in the previous section. Then, we apply an LQG control for three degrees of freedom: $x_{f}$*,*
$y_{f}$ and $\psi_{f}$, in the DP system of the Figure 1.

Figure 4 shows the structure of the LQG controller. It must be noted that it includes a PI controller in order to compensate the drift effects induced by the second-order wave effect previously described. The controller calculates the vector of forces and moments $F_{c}\left(t\right)\;=\;{\left[X_{c},Y_{c},N_{c}\right]}^{T}$ by means of the current position of the caisson and the reference vector $Ref\left(t\right)\;=\;{\left[x_{ref},y_{ref},\psi_{ref}\right]}^{T}$.

The implementation of the LQR regulator was carried out with the model in state space of Equations (3) and (4).

The cost function is stablished by doing [3]:(23)$$J\left(u\right)\;=\;\int_{0}^{\infty }\left[x_{L}{}^{T}Q_{LQR}x_{L}+u_{L}{}^{T}R_{LQR}u_{L}\right]dt,$$
where $Q_{LQR}\in R^{\mathrm{nxn}}$ is a negative undefined matrix and $R_{LQR}\in R^{\mathrm{pxp}}$ is positive definite. 

The linear feedback of the control state is calculated by minimizing cost function, Equation (23), being:(24)$$u_{LQR}\;=\;-K_{LQR\;}x_{f}$$
where:(25)$$K_{LQR}\;=\;R_{LQR}{}^{-1}B_{LQR}{}^{T}P_{LQR}.$$

To do this, it is necessary to calculate $P_{LQR}$, which is the positive and symmetric definite solution of the Riccati algebraic equation:(26)$$A_{L}{}^{T}P_{LQR}\;+\;P_{LQR}A_{L}\;-\;P_{LQR}B_{L}R_{LQR}{}^{-1}B_{L}{}^{T}P_{LQR}\;+\;Q_{LQR}\;=\;0.$$

## 5. Control Allocation

The deployment procedure includes eight winches that are the actuators of the system. See Appendix A for more details relative to the winches and mooring lines configuration. These actuators act by means of cables attached to the caisson. Therefore, winches exert different tensions that result in different forces and moments on the caisson.

The control allocation calculates the tensions that each winch must exert on the caisson with the objective to obtain as a result of these actions the forces commanded by the dynamic control in the three control variables already mentioned.
$$T_{cbw}\;=\;[T_{cbw1},T_{cbw2},T_{cbw3},T_{cbw4},T_{cbw5},T_{cbw6},T_{cbw7},T_{cbw8}].$$

The algorithm of the control allocation is the following (Algorithm 1) [19]
**Algorithm 1** The algorithm of the control allocation*Initialize*Tmax*= 10 and *nl*= 8;**Load the control signals [*Xc*,* Yc***,*** Nc***];****for*k (∈1,…,∞)*do**if X > 0 then*
      
$f_{12}=A_{c1}$
*;*

*else*

* if X < 0 then*

       
$f_{12}=A_{c2}$
   * end if*
*end if*
*if Y > 0 then*

         
$f_{34}=A_{c3}$
*;*
*else*
   * if Y < 0 then*

        
$f_{34}=A_{c4}$
***;***
   * end if*
*end if*
*if N > 0 then*

          
$f_{56}=A_{c5}$
***;***

*else*

* if N < 0 then*

          
$f_{56}=A_{c6}$
***;***

*    end if*

*end if*

*Add the contributions of all of the axis*
 $S={\left[f_{12}+f_{34}+f_{56}\right]}^{T}$
***;***
*Adjust the gain* 
$S=\frac{S}{2}\;$
*;*
*Calculate the tensions of the*
$n_{l}$
***lines***
$T_{c}=S*abs\left(\left[y_{fx},y_{fy},y_{fN}\right]\right)$
*;*
*for*
$j\;\left(\in 1,...,n_{l}\right)$ 
*do*
   * if T(j) >*
$T_{max\;}$
*then*
      
* T(j) =*
$\;T_{max\;}$
*;*
*    end if*
*end*
*end*


$T_{\max}$ maximum value of the tension, $n_{l}$ number of anchor lines.

The matrices of the Algorithm are the following:$$A_{c1}=\left[\begin{array}{ccc} 0 & 0 & 0\\ 1 & 0 & 0\\ 0 & 0 & 0\\ 1 & 0 & 0\\ 0 & 0 & 0\\ 0 & 0 & 0\\ 0 & 0 & 0\\ 0 & 0 & 0 \end{array}\right]A_{c2}=\left[\begin{array}{ccc} 1 & 0 & 0\\ 0 & 0 & 0\\ 1 & 0 & 0\\ 0 & 0 & 0\\ 0 & 0 & 0\\ 0 & 0 & 0\\ 0 & 0 & 0\\ 0 & 0 & 0 \end{array}\right]A_{c3}=\left[\begin{array}{ccc} 0 & 0 & 0\\ 0 & 0 & 0\\ 0 & 0 & 0\\ 0 & 0 & 0\\ 0 & 1 & 0\\ 0 & 1 & 0\\ 0 & 0 & 0\\ 0 & 0 & 0 \end{array}\right]$$
$$A_{c4}=\left[\begin{array}{ccc} 0 & 0 & 0\\ 0 & 0 & 0\\ 0 & 0 & 0\\ 0 & 0 & 0\\ 0 & 0 & 0\\ 0 & 0 & 0\\ 0 & 1 & 0\\ 0 & 1 & 0 \end{array}\right]A_{c5}=\left[\begin{array}{ccc} 0 & 0 & 0\\ 0 & 0 & 1\\ 0 & 0 & 1\\ 0 & 0 & 0\\ 0 & 0 & 1\\ 0 & 0 & 0\\ 0 & 1 & 0\\ 0 & 1 & 1 \end{array}\right]A_{c6}=\left[\begin{array}{ccc} 0 & 0 & 1\\ 0 & 0 & 0\\ 0 & 0 & 0\\ 0 & 0 & 1\\ 0 & 0 & 0\\ 0 & 0 & 1\\ 0 & 1 & 1\\ 0 & 1 & 0 \end{array}\right]$$


The tensions $T_{cbw}$ are multiplied for gain of the winches and then, forces and moments are provoked in the center of gravity of the caissons, for details relative to this aspect see [19].

## 6. Simulation Results

The simulations performed in this section are done to verify the behavior of the LQG controller implemented in the DP system shown in Figure 1. All of them are referred to the caisson, whose main particulars are summarized in the Appendix A. The simulations were carried out in the Matlab-Simulink environment with a time step of 0.1 s. These simulations are divided into two groups: in the former, the DP system of Figure 1 with the LQG controller is simulated and it is compared to the results given by a classical controller, see [19] for more details relative to the controller. In the latter, a Monte Carlo study of 200 realizations is developed, where the performance of the LQG controller is compared to the one provided by the classical controller, both of them in the DP system of Figure 1.

In the simulation, the reference vector was set to *Ref*(*t*) *=*
${\left[5m,\;4m,\;0.175rad\right]}^{T}$. The KF matrices were tuned by doing:



$Q=diag\left(\left[0.001,1,0.001,1,0.001,1,1\times 10^{12},1,1\times 10^{12},1,1\times 10^{15},1\times 10^{12}\right]\right).$





$R=diag\left(\left[1\times 10^{-3},1\times 10^{-3},1\times 10^{-3},1\times 10^{-3},1\times 10^{-3},1\times 10^{-3}\right]\right).$

The LQR matrices were tuned by doing:



$Q_{LQR}=[4\times 10^{6},0,0,0,0,0;0,4\times 10^{6},0,0,0,0;0,0,4\times 10^{6},0,0,0;0,0,0,4\times 10^{6},0,0;$



$0,0,0,0,4\times 10^{6},0;0,0,0,0,0,4\times 10^{6}].$





$R_{LQR}=\left[12\times 10^{4},0,0;0,12\times 10^{4},0;0,0,8\times 10^{5}\right].$



The wave parameters for all the simulations are σ = 0.125, $\omega_{0}$
*=*
1.2 and λ=0.1, see Figure 5.

The DP system with the LQG controller counteract the second-order drift effects. This counteracting is due to the PI controller included in the first control loop shown in the Figure 4.

In addition, the implemented system follows the references trajectory (a line between the initial and final point, see Figure 6. It can be observed that the response is stable and has no steady state error. Furthermore, the first-order wave effects were filtered by KF implemented in the system. As a result of that, the controller receives the signals free of oscillations and the levels of noise corresponding to the standard instrumentation used on board the caisson. This can be seen in the zoomed part of Figure 7. Moreover, the actuator signals are not saturated. This means that oscillations are quite limited, see Figure 8. In the same way, the tension signal commanded to the winches does not present saturations or excessive oscillations, see Figure 9.

Taking into account the results, the system is able to perform the dynamic positioning of the caisson with high precision, compensating and filtering the effects of the environment in which the operations will take place. 

Figure 10 expose the response of the DP system with the classical controller previously cited with a first-order network as a filter. By comparing Figure 6 and Figure 10, it can be seen a more elevated oscillation in the response of the later one. Moreover, a sharp behavior of controller, seen in Figure 11, represents a significant increment of the control efforts, even with saturations with respect to Figure 8. It can be concluded that the LQG control strategy brings out a controller performance closer to their optimal operating range. This will extend the service life of the winches and contribute to a safely maneuvers development.

Finally, a Monte Carlo study was carried out for dynamic position of the caisson at the location x = 0 and y = 0. It is concluded in Figure 12 that the performance of the DP system with the classical controller (Figure 12b) clearly provides a higher dispersion of the position of the caisson compared to the LQG controller (Figure 12a).

## 7. Conclusions

In this paper, based on a KF, an LQG control strategy has been applied to for dynamic positioning of floating moored concrete caissons. The simulation results evidenced a good compensation of the second-order wave drift motions thanks to the PI controller included the LQG control loop. Additionally, the low oscillation in the performance of the caisson’s motion also indicates a good compensation of the first-order wave induced motions.

Furthermore, the results given by the classical approaches to the ones provided by the LQG control strategy were compared. From this comparison it can be concluded that the LQG control strategy provides a reduction in the oscillations. Moreover, it also contributes to reducing in the control efforts, avoiding potential saturations. Therefore, this may lead to longer life cycles of winches and actuators, as well as to a safer performance of the manor lines. Finally, the Monte Carlo results shows that the LQG control strategy present a significant lower dispersion for station keeping maneuvers compared to the classical controller. 

As it is well-known, the application of the LQG control method gives good results if the dynamic behavior of the system is linear. As previously explained, due to the low speed at which the operations are performed, the linear approximation can properly represent the dynamics of the caisson. However, if these operational conditions significantly change, the linear approximation might not correctly represent the behavior of the caisson and the performance of the controller may be degraded.

## Figures and Tables

**Figure 1 sensors-21-06496-f001:**
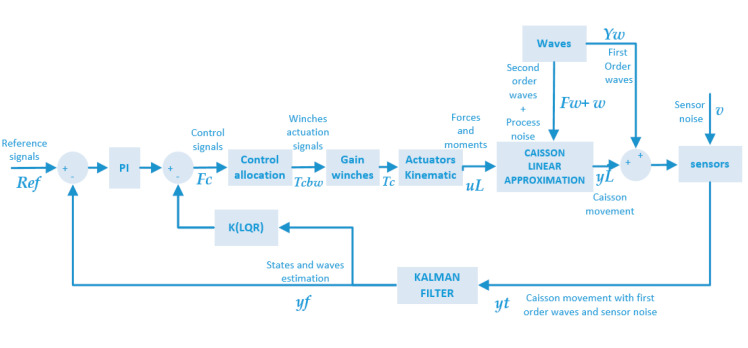
LQG control loop for dynamic positioning of caissons based on the KF.

**Figure 2 sensors-21-06496-f002:**
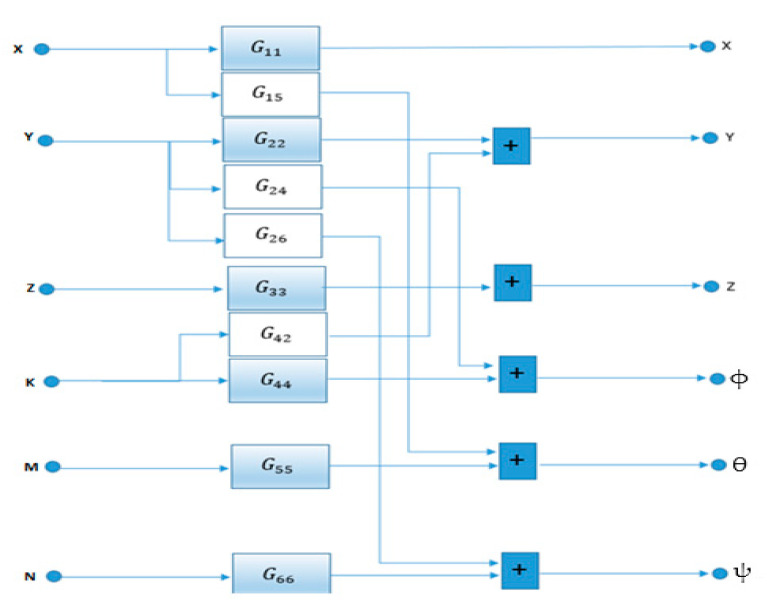
Structure of the linear approximation for a caissons’ draft of 10.75 m.

**Figure 3 sensors-21-06496-f003:**
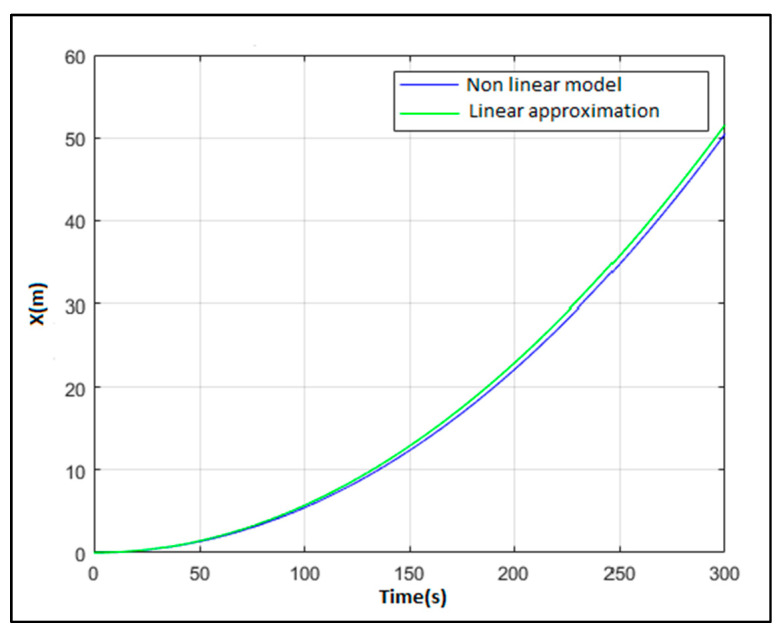
Validation of the linear approximation for the surge degree of freedom. Step input = 10,000 N.

**Figure 4 sensors-21-06496-f004:**
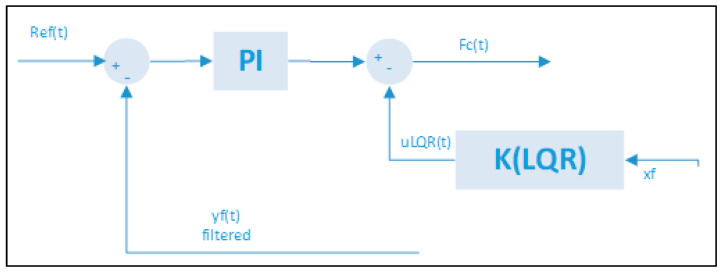
LQG control strategy for three degrees of freedom x, y and ψ.

**Figure 5 sensors-21-06496-f005:**
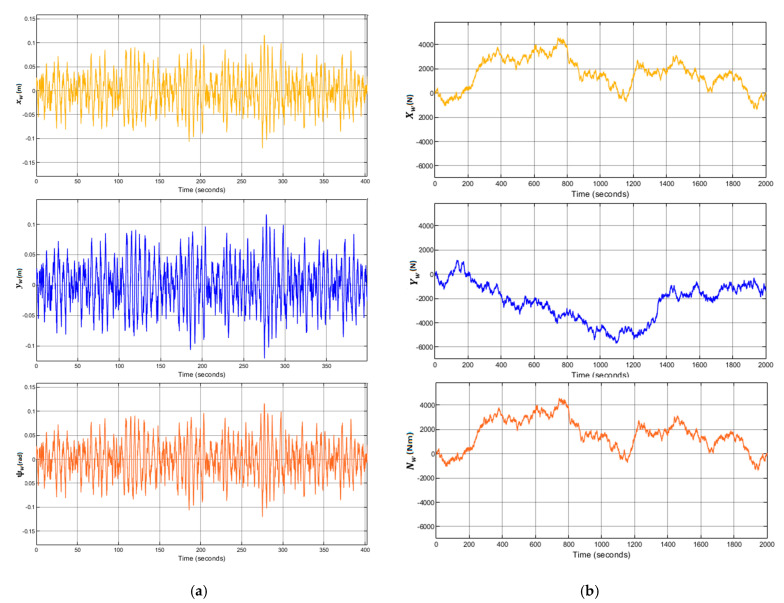
(**a**) Systems with LQG controller control results with waves. First-order waves forces and moments applied to the caisson. (**b**) Second-order waves forces and moments applied to the caisson.

**Figure 6 sensors-21-06496-f006:**
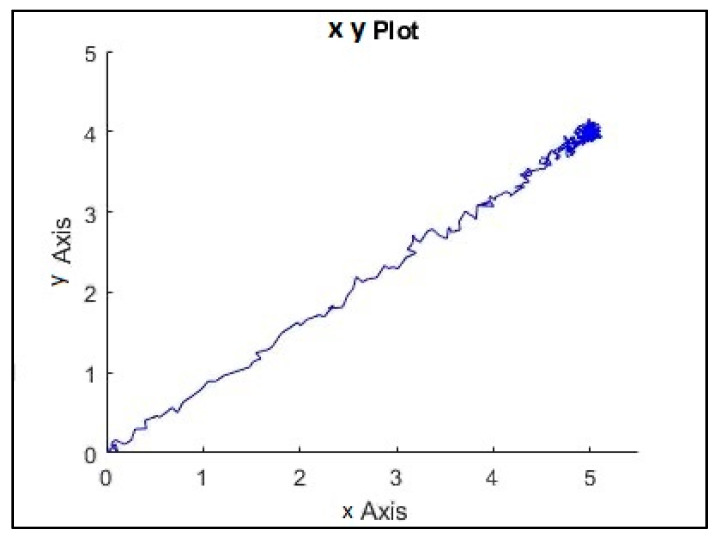
System with LQG controller control results with waves, positions (x,y).

**Figure 7 sensors-21-06496-f007:**
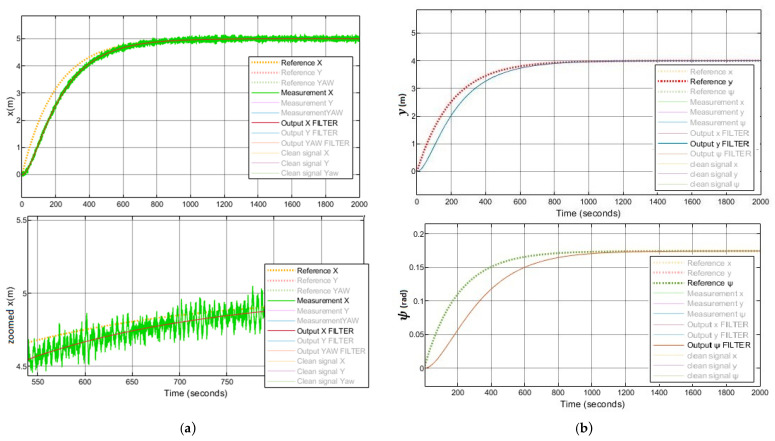
(**a**) System with LQG controller control results with waves, position x. (**b**) System with LQG controller control results with waves, position y and ψ.

**Figure 8 sensors-21-06496-f008:**
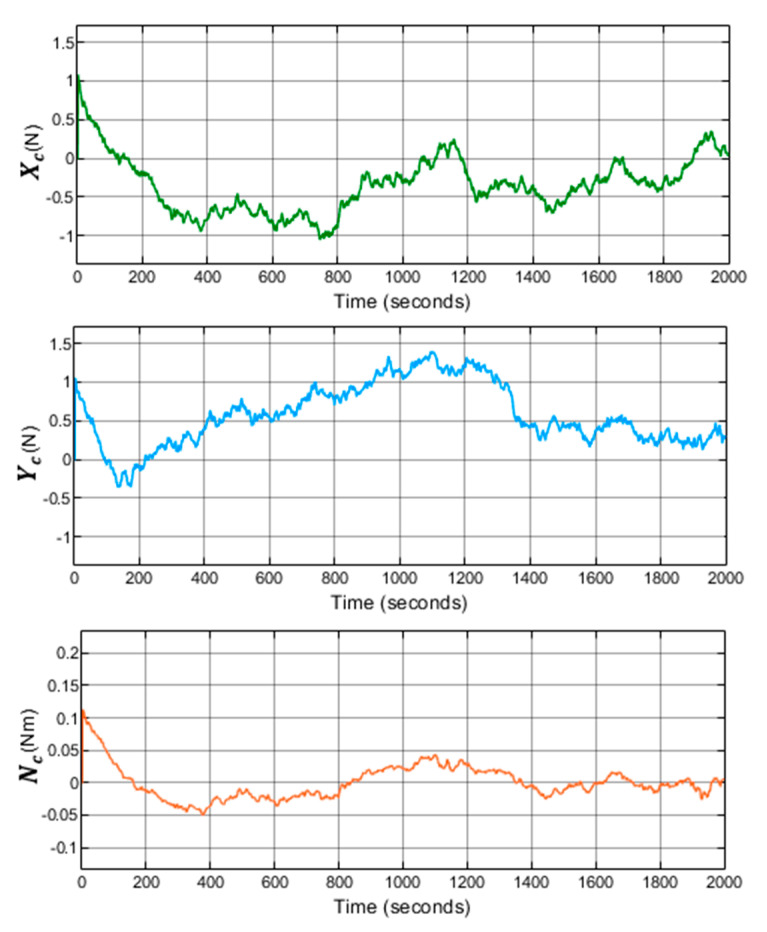
System with LQG controller control results with waves. Control signals calculated by the double-loop controller.

**Figure 9 sensors-21-06496-f009:**
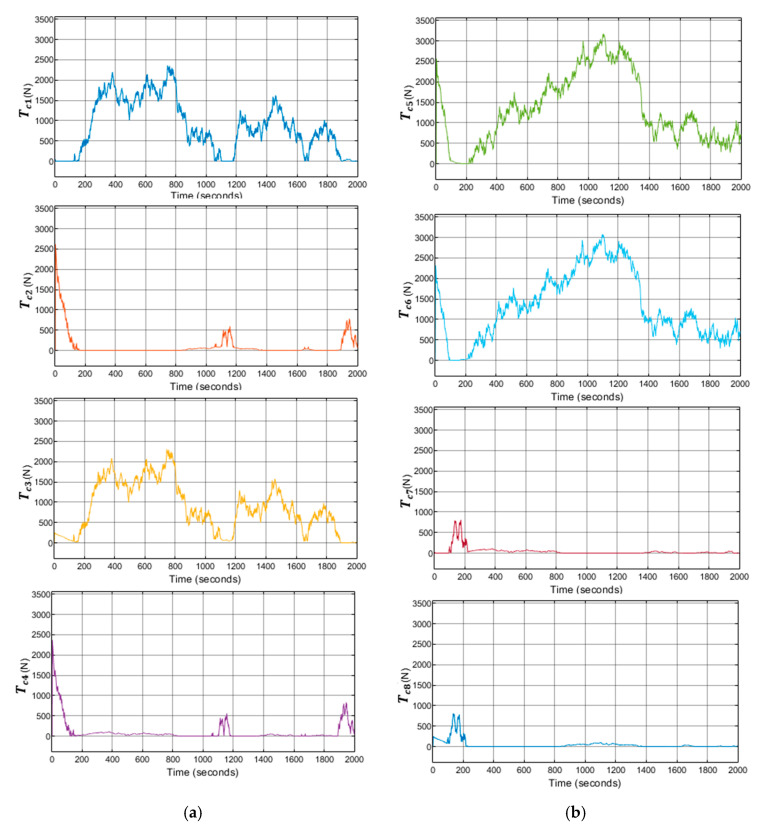
(**a**) System with LQG controller control results with waves, tensions applied in each of the winches 1–4. (**b**) System with LQG controller control results with waves, tensions applied in each of the winches 5–8.

**Figure 10 sensors-21-06496-f010:**
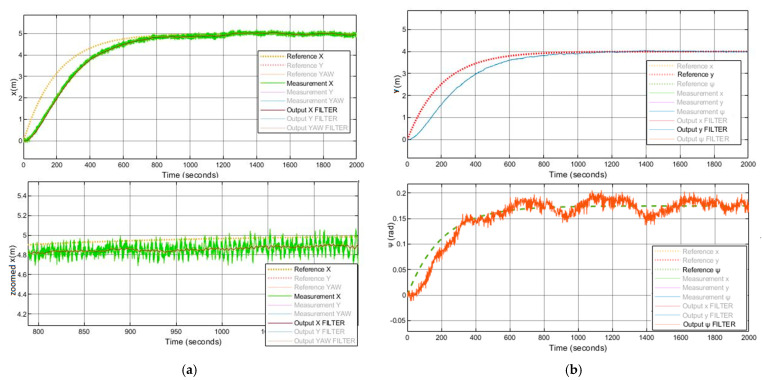
(**a**) System with classical controller control results with waves, position x. (**b**) System with classical controller control results with waves, position y and ψ.

**Figure 11 sensors-21-06496-f011:**
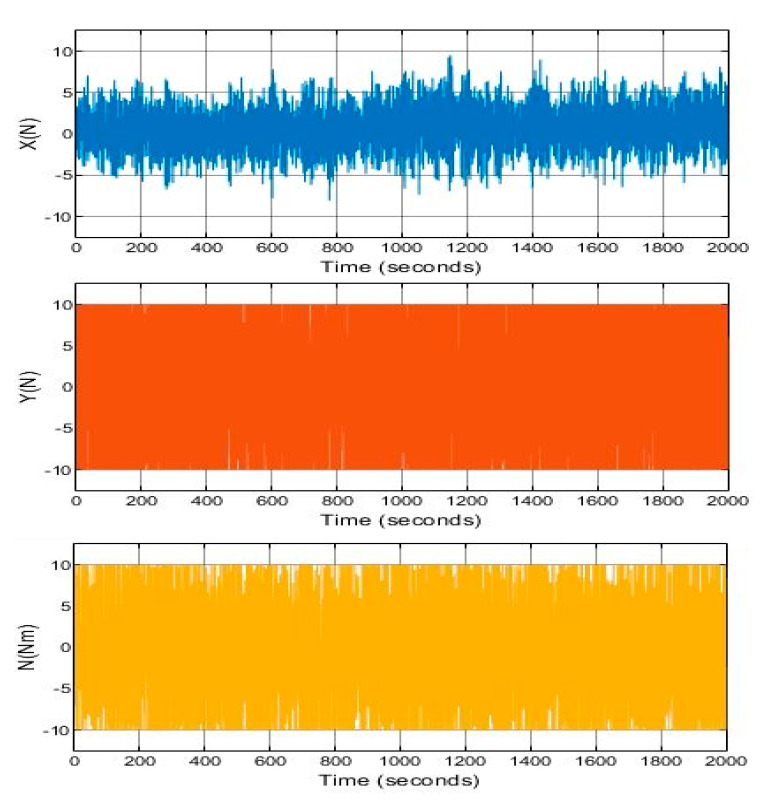
System with classical controller control result with waves. Force in *X*, *Y* and *N*.

**Figure 12 sensors-21-06496-f012:**
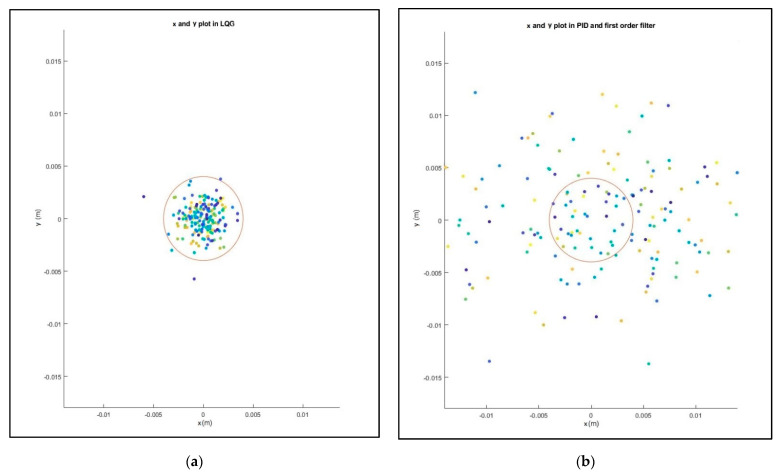
(**a**) System with LQG controller control result with waves. Monte Carlo study of 200 realizations. (**b**) System with classical controller control result with waves. Monte Carlo study of 200 realizations.

**Table 1 sensors-21-06496-t001:** Family of transfer functions of the linear approximation.

Name	Transfer Function Aproximation
** $G_{11}$ **	$\frac{a_{n11}}{s^{2}}$
** $G_{15}$ **	$\frac{a_{n15}s^{2}+b_{n15}s+c_{n15}}{a_{d15}s^{3}+b_{d15}s^{2}+c_{d15}s}$
** $G_{22}$ **	$\frac{a_{n22}}{s^{2}}$
** $G_{24}$ **	$\frac{a_{n24}s^{2}+b_{n24}s+c_{n24}}{a_{d24}s^{3}+b_{d24}s^{2}+c_{d24}s}$
** $G_{26}$ **	$\frac{a_{n26}}{s^{2}}$
** $G_{33}$ **	$\frac{a_{n33}}{a_{d33}s^{2}+b_{d33}s+c_{d33}}$
** $G_{42}$ **	$\frac{a_{n42}s^{2}+b_{n42}s+c_{n42}}{a_{d42}s^{3}+b_{d42}s^{2}+c_{d42}s}$
** $G_{44}$ **	$a_{44}s^{2}+b_{44}s+c_{44}$
** $G_{55}$ **	$\frac{a_{n55}}{a_{d55}s^{2}+b_{d55}s+c_{d55}}$
** $G_{66}$ **	$\frac{a_{n66}}{s^{2}}$

## Data Availability

Not applicable.

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
