# Peer review of "LQG Control for Dynamic Positioning of Floating Caissons Based on the Kalman Filter"

_sensors, 2021, doi:10.3390/s21196496_

Round 1
Reviewer 1 Report
After making revisions based on the reviewers’ comments, this paper summarized the existing key research, explained the limitations of the existing methods and the innovation of this research, and analyzed the simulation results in detail. But there are still some details in this article that need to be modified before publishing:
(1) Figure 2 appears three times in the text, please correct it;
(2) Figure 3 appears twice in the text, please correct it. And Figure 3 is not clear, it is recommended to improve its clarity;
(3) Figure 3, exhibits the fit of the obtained linear approximation for the surge degree of freedom with respect to the data generated by the nonlinear model of equation (1). Can the effect of fitting be described accordingly?
Author Response
See the attached pdf.

Reviewer 2 Report
Line 59 - it would be better if authors could briefly explain the contribution of the literature mentioned instead of citing 20 different sources "which apply a wide variety of control methods". Which are the existing methods, and how do they differ from the one proposed in this paper?
Lines 367 through 372 - this paragraph introduces the limitations of the proposed model. It is very interesting to know under which circumstances could the system be degraded; when the model is not performing ideally. I would like to see this idea developed in a discussion section (preferably before 8. Conclusions), where the different aspects (limitations) that could interfere would be considered.
Author Response
See the attached pdf.
